# Changes in historical typhoid transmission across 16 U.S. cities, 1889-1931: Quantifying the impact of investments in water and sewer infrastructures

**Maile T. Phillips** [1] *, **Katharine A. Owers** [1], **Bryan T. Grenfell** [2], **Virginia E. Pitzer** [1]

**1** Department of Epidemiology of Microbial Diseases, Yale School of Public Health, New Haven, Connecticut, United States of America, **2** Department of Ecology and Evolutionary Biology, Princeton University, Princeton, New Jersey, United States of America

* maile.phillips@yale.edu

**Data Availability Statement:** Data and code for this manuscript are available online at https://

## Abstract

Investments in water and sanitation systems are believed to have led to the decline in typhoid fever in developed countries, such that most cases now occur in regions lacking adequate clean water and sanitation. Exploring seasonal and long-term patterns in historical typhoid mortality in the United States can offer deeper understanding of disease drivers. We fit modified Time-series Susceptible-Infectious-Recovered models to city-level weekly mortality counts to estimate seasonal and long-term typhoid transmission. We examined seasonal transmission separately by city and aggregated by water source. Typhoid transmission peaked in late summer/early fall. Seasonality varied by water source, with the greatest variation occurring in cities with reservoirs. We then fit hierarchical regression models to measure associations between long-term transmission and annual financial investments in water and sewer systems. Overall historical $1 per capita ($16.13 in 2017) investments in the water supply were associated with approximately 5% (95% confidence interval: 3–6%) decreases in typhoid transmission, while $1 increases in the overall sewer system investments were associated with estimated 6% (95% confidence interval: 4–9%) decreases. Our findings aid in the understanding of typhoid transmission dynamics and potential impacts of water and sanitation improvements, and can inform cost-effectiveness analyses of interventions to reduce the typhoid burden.

## Author summary

Typhoid fever remains a major source of morbidity and mortality in low- and middle-income countries. Historical investments in water and sanitation systems are thought to have led to the decline in typhoid fever in developed countries, such that most of the global burden of disease now occurs in regions with poor sanitary conditions and inadequate access to clean water and sanitation. However, there is limited empirical evidence to quantify the impact of investments in water and sanitation on typhoid fever incidence.

github.com/mailephillips/Historical-Typhoid. The original typhoid death data are also freely available at https://www.tycho.pitt.edu/data/.

**Funding:** This work was supported by the Bill and Melinda Gates Foundation (OPP1116967, OPP1151153 to VEP; OPP1091919 to BTG; URL: https://www.gatesfoundation.org/) and the Wellcome Trust (Strategic Award 106158/Z/14/Z; URL: https://wellcome.ac.uk/). The funders had no role in study design, data collection and analysis, decision to publish, or preparation of the manuscript.

**Competing interests:** The authors have declared that no competing interests exist.

We developed a mathematical model to examine trends in weekly typhoid mortality data from 1889–1931 in 16 U.S. cities. Through this analysis, we were able to examine how seasonal patterns of typhoid transmission varied geographically and historically depending on the water supply and treatment, and quantify the relationship between investments in water and sanitation infrastructures and long-term typhoid transmission rates. Our findings have important implications for the understanding of typhoid transmission dynamics and potential impact of improvements in water and sanitation infrastructure. Resource-poor countries must prioritize spending on public health issues, weighing the costs and benefits of interventions. Our results can help to inform comparative cost-effectiveness analyses of different interventions to reduce the global burden of typhoid fever.

## Introduction

Typhoid fever is caused by infection with the bacteria *Salmonella enterica* serovar Typhi, which is mainly transmitted through fecal contamination of food or water [1]. In many developed countries, including the United States (U.S.), investments in water and sewer infrastructures led to the decline in typhoid incidence in the beginning of the 20th century, such that the majority of the global burden now occurs in countries where sanitary conditions are poor and access to clean water and sanitation is lacking [1–4].

Examining short- and long-term trends in typhoid incidence can provide insights into factors driving transmission [5]. In many countries, typhoid fever follows a seasonal pattern, with peak incidence occurring around the same time every year [6, 7]. Seasonality in typhoid exhibits distinct patterns by region and latitude, and can be influenced by rainfall, temperature, and other climatic factors [7]. However, drivers of seasonal patterns in typhoid are not yet fully understood.

Long-term patterns in typhoid cases have also been investigated, particularly in countries where cases have declined to almost zero [5]. In the U.S., the number of typhoid deaths decreased from a reported 35,000 in 1900 to three from 1999–2006 despite a 4.3-fold population increase [8–11]. While it is commonly accepted that investments in water and sanitation are responsible for the decline in typhoid fever, there is limited empirical evidence to support this claim. In one study, Cutler and Miller found that the introduction of clean water technologies was responsible for almost half of the mortality reduction in major cities at the beginning of the 20th century; however, they did not consider complexities of the disease transmission process [12].

In this study, we developed mathematical and statistical models to examine seasonal and long-term trends in typhoid transmission from 1889–1931 in 16 U.S. cities. Our objectives were two-fold: (1) to examine how seasonal patterns of typhoid transmission varied geographically and historically depending on the water supply and treatment; and (2) to quantify the relationship between investments in water and sanitation infrastructures and long-term typhoid transmission rates.

## Methods

### Study design, data, and variables

We extracted reported weekly typhoid mortality from 1889 to 1931 at the city level from the Project Tycho database [13, 14]. Cities were chosen based on two criteria: (1) at least 1,000 typhoid deaths were reported during the study period, and (2) less than 25% of weekly data

 

was missing. These exclusion criteria resulted in data for 16 U.S. cities (S1 Fig). While errors in disease diagnosis and missing data make underreporting likely, the consistency of reporting over time allows for our analysis [14, 15].

Yearly population estimates were obtained from the U.S. Census Bureau [16, 17]. The population <1 year of age was used as a proxy for births, since birth rate data was not available and typhoid is rare in <1-year-olds [18]. New York City population estimates were adjusted for the consolidation of the five boroughs (including Brooklyn) in 1898 [19]. We also accounted for this change by multiplying the number of reported typhoid deaths in Brooklyn by a factor of 1.28 (i.e. the relative population size of the other boroughs, for which we did not have separate mortality data) and adding this to typhoid mortality data from New York City (previously only Manhattan). For all cities, cubic splines were used to extrapolate weekly population estimates (S2 Fig).

Financial data on water supply and sewer systems for each city were extracted from U.S. Census Bureau yearly reports [16]. We obtained data on water and sewer systems across five categories: "receipts," "expenses," "outlays," "value," and "funded debt", and used the first three to estimate the "overall investment" (Table 1). The main financial variable of interest, "overall investment", represents the cumulative per capita investment in water supply and sewer systems. It was calculated as the sum of the annual acquisition/construction costs (cumulative outlays) and maintenance/operation costs (expenses) after subtracting yearly receipts (Table 1). All variables were adjusted yearly for inflation to 1931 dollars using the Bureau of Labor Statistics' Consumer Price Index [20], then divided by the yearly city population to generate per capita estimates. Data on specific water supply interventions for each city were extracted from a variety of sources (S1 Table) [17].

All cities had missing data on weekly typhoid mortality, due to the nature of the historical data. In many cases, missing mortality counts were instances of zero cases, because cities

**Table 1. Definitions of financial variables.** Each of the six categories of financial variables used in this study are described, as defined by the U.S. Census Bureau in its annual "Financial Statistics" series (the source of these variables).

| | Description |
|---|---|
| **Maintenance and operation** | |
| *Receipts* | Receipts for payments for governmental costs. These receipts usually take the form of money, bills receivable, land, and services. All city revenue receipts were recorded in the city books for municipally-operated water supply and sewer systems for the public or city (excluding interest from current deposits). |
| *Expenses* | City government costs, other than interest, of (1) services employed, property rented, and materials consumed in connection with maintenance and operation; (2) losses from deflation, bank failures, and related causes; and (3) depreciation of permanent properties and public improvements. |
| **Acquisition and construction** | |
| *Outlays* | Total annual amounts paid by the city for the acquisition or construction of permanent lands, properties and public improvements. These include payments for additions made to previously acquired or constructed properties. |
| **Value and debt** | |
| *Value* | Total estimated value of the public properties (including depreciation), including both the business value and the physical value of the building and equipment. This amount is estimated separately by city officials, and is acknowledged to not be estimated uniformly across cities. |
| *Funded debt* | Long-term debts or debt liabilities in the form of bonds or certificates of indebtedness that the city government is under obligation to pay. |
| **Cumulative investment** | |
| *Overall investment* | Overall investment in the water supply or sewer systems, defined as the cumulative sum of the amount spent each year on acquisition/construction (outlays) and maintenance/operation (expenses minus receipts) of water or sewer infrastructure. |

frequently only reported during weeks when deaths occurred. To account for both true zero counts and missing data, mortality data were coded as zeroes if there were fewer than 13 consecutive weeks of missing death counts, and imputed as missing data if there were 13 or more consecutive weeks. We imputed missing data using the package "imputeTS" in R[21], performing Kalman smoothing (function *na_kalman*) to preserve the seasonality and overall trends of the time series (S3 Fig). This package and algorithm are commonly used for univariate time series imputation. We conducted a sensitivity analysis to assess how this arbitrary 13-week cut-off could impact our results (S2 Text).

After imputation, weekly typhoid mortality counts and population estimates were aggregated into four-week periods to approximate the generation interval of typhoid [22, 23]. The generation interval can be defined as the time between when an infector is infected and when an individual is infected by that the infector [23, 24]. In this study, the generation interval was based on data from the natural history of typhoid infection, derived from human challenge studies. Other studies suggest that TSIR models are not overly sensitive to having a precise estimate for the generation interval [25]. Since the mortality data were later log-transformed, we added one to every four-week data point before adjusting for underreporting and before fitting the model; a sensitivity analysis was again performed to assess the impact of adding different values.

## Statistical methods

We conducted preliminary analyses to describe differences in typhoid mortality trends between cities and pre- to post-intervention. First, we fit generalized linear models (GLMs) with linear time trends and one-year and six-month harmonics to the pre- and post-intervention time series (defined as two years after the first water supply intervention) for each city. The "first" intervention is defined as the initial occurrence of a municipally-reported method or process that aimed to improve the water quality in a city's main water source, and was used only in the preliminary analyses to define the pre- and post- intervention period. The six- and 12-month harmonics allow for an overall annual variation plus additional fluctuations, if any; these were identified using Fourier and wavelet analyses. We compared intercepts, slopes, and six-month and one-year amplitudes for the pre- and post- periods in the GLMs, and plotted the overall six- and 12-month amplitudes on a map of the U.S. to examine spatial patterns.

We then fit Time-series Susceptible-Infectious-Recovered (TSIR) models [26] to each city's pre- and post-intervention time series to investigate seasonal and long-terms trends in typhoid transmission rates. TSIR models are a well-established approach to examine associations between external variables and infectious disease transmission rates by conditioning on the susceptible population and exposure to a pathogen to extract rates of infectiousness inferred from the time series [27]. These models estimate the disease transmission rate by reconstructing the underlying susceptible and infectious populations. This method explicitly attributes autocorrelation in the data to the interaction between susceptible and infectious individuals.

In general, new infections at time $t+1$ ($I_{t+1}$) arise from transmission from infectious ($I_t$) to susceptible ($S_t$) individuals at time $t$:

$$I_{t+1} = \beta_t I_t^\alpha S_t \qquad (1)$$

where $\beta_t$ is the disease transmission rate at time $t$. The exponent $\alpha$ allows for heterogeneous population mixing and corrects for discretization of the continuous-time infection process [28].

We modified Eq 1 to account for the unique features of typhoid epidemiology, including the contribution of chronic carriers ($C$) to the prevalence of infection. Furthermore, we

separated the transmission parameter $\beta_t$ into seasonal and long-term components ($\beta_{seas}$ and $\beta_{lt}$, respectively). Thus, the TSIR model for typhoid is as follows:

$$I_{t+1} = \beta_{lt}\beta_{seas,j}(I_t + C)^\alpha S_t \qquad (2)$$

where $\beta_{seas,j}$ reflects the annual seasonally varying transmission parameter ($j$ = 1,2,. . .13 for the number of distinct four-week generation intervals in one year), and $\beta_{lt}$ (558 distinct values for the number of generation intervals over the 43-year period, minus 1 for the reconstruction of $I_{t+1}$) captures trends and any seasonal variation lasting longer than one year. We fixed $\beta_{seas,13}$ = 1 and estimated the remaining $j$ = 1,2,. . .,12 seasonal transmission parameter in comparison to the thirteenth month. We estimated $\beta_{lt}$ using a semi-parametric method described below and in more detail in the S1 Text.

Eq 2 can then be log-transformed:

$$log(I_{t+1}) = log(\beta_{lt}) + log(\beta_{seas,j}) + \alpha \, log(I_t + C) + log(S_t). \qquad (3)$$

The TSIR equation is now on the additive scale, and can be incorporated into regression frameworks (S1 Text). This method has been explained in detail elsewhere [26].

With the goal of extracting the seasonal and long-term transmission rates ($\beta_{lt}$ and $\beta_{seas,j}$), we needed to first reconstruct the susceptible, infectious, and chronic carrier populations. We estimated some of these terms differently for our exploratory and main analyses, but both analyses utilized regression and maximum likelihood estimation to infer these terms from the disease and census data.

The susceptible population at time $t$ is equal to the previous susceptible population plus new births minus new infections, summarized as follows:

$$S_t = \bar{S} + D_0 + \sum_{k=0}^{t-1}b_k - \sum_{k=0}^{t-1}I_k \qquad (4)$$

where $\bar{S}$ is the mean susceptible population over the study period, $D_0$ is the deviation of the susceptible population from the mean at time zero, $\sum_{k=0}^{t-1} b_k$ is the sum of births up to time $t$, $\sum_{k=0}^{t-1} I_k$ is the sum of "true" infections up to (but not including) time $t$, and $k$ denotes the time point ranging from the beginning of the study up until just before time $t$. The number of "true" infections at time $t$ ($I_t$) is estimated from the observed deaths at time $t$ ($Y_t$) divided by the underreporting fraction ($\rho$), which in this case also accounts for the case fatality rate. Eq 4 can be rearranged as

$$\sum_{k=0}^{t}b_k = \left(\frac{1}{\rho}\right)\sum_{k=0}^{t}Y_k - D_0 + D_t \qquad (5)$$

to estimate the underreporting fraction (slope), deviation at time zero (intercept), and model residuals ($D_t = S_t - \bar{S}$) using linear regression. We used only the first ten years of typhoid mortality and census data (prior to the introduction of water and sanitation interventions) [25, 26] to estimate the rate of underreporting of infectious individuals, and assumed that $\rho$ remained constant over the entire 43-year study period.

To estimate $C$ and $S_t$ ($= D_t + \bar{S}$) in the preliminary analysis, we maximized the likelihood of the fitted regression (Eq 5) over different values of $C$ and $\bar{S}$, each ranging from 0 to the maximum population size over the time period. For the preliminary analysis, we then fit Eq 3 using ordinary least squares regression.

For our main analysis, we used the same estimates for the infectious population (adjusted for underreporting) and chronic carriers from the preliminary analysis, but modified the calculation for the susceptible population to include waning of immunity. Instead of using the

residuals from Eq 5, we modelled the susceptible population at time $t$ as a function of the total population at time $t$ minus the previously infectious and recovered individuals:

$$S_t = N_t - \sum_{i=0}^{m} I_{t-i} \kappa_i \qquad (6)$$

where $\kappa_i$ is the degree of immunity $i$ generation intervals after infection.

Once we had estimates for the susceptible, infectious, and chronic carrier components of Eq 3, we fit the model via weighted least squares regression using a range of values for smoothing and spline penalty parameters. For the final model, we chose the one with the smoothing and spline penalty parameters that resulted in the lowest sum of squared differences between each point and its out-of-sample prediction over all points.

The model-fitting process is described in detail in the S1 Text; additional details about TSIR models can be found elsewhere [26, 29, 30]. We performed sensitivity analyses on the various components of the model, as described in the S2 Text.

## Examining predictors of seasonal and long-terms trends in transmission

Once we fit the optimal TSIR model for each city, we extracted the seasonal and long-term transmission rates. Seasonal transmission parameters were plotted separately for each city and aggregated by water source type. We calculated the mean estimate (among all cities and across water source types) in each month. Months were considered to have significantly low or high seasonal transmission if their confidence intervals were entirely below or above one, respectively. The percentage of cities with seasonal transmission significantly below or above one in each month were calculated overall and by water source type.

To examine associations between long-term typhoid transmission and financial investments in water and sewer systems, we fit hierarchical regression models for each financial variable separately. We fit several variable transformations and model formulations and chose a linear model with a log-transformed outcome following exploratory analyses. The final approach has fixed and varying city-level intercepts and slopes:

$$\log(\beta_{lt,i,t}) = (d_0 + \delta_{0,i}) + (d_1 + \delta_{1,i}) X_{i,t} \qquad (7)$$

where fixed intercept $d_0$ is the average log-transmission rate of typhoid across cities with no investments in water and sanitation, random intercept $\delta_{0,i}$ represents the deviation from the fixed intercept for city $i$, fixed slope $d_1$ is the average change in log-transformed typhoid transmission across cities for a \$1 per capita increase in the financial variable, random slope $\delta_i$ is the deviation from the fixed slope for city $i$, and $X_{i,t}$ is the financial investment for city $i$ in year $t$.

Missing financial variable data were assumed to be missing completely at random and were omitted from analyses. Due to multicollinearity between most of the financial variables, it was not possible to fit regression models with multiple predictors. However, the main variables of interest, overall investments in the water supply and sewer systems, provide a representation of cumulative financial investments as a whole over the time period.

## Model validation

To validate the TSIR models and assess their predictive ability, we went back and fit each TSIR model to the first 38 years of data (1889–1926). Using the fitted model parameters, we projected forward for the last five years (1927–1931) and compared the observed and predicted typhoid mortality. To predict the long-term typhoid transmission rate, we used the relationship with overall investment in the water supply identified by the hierarchical regression analysis. This variable had the highest marginal and conditional $R^2$ among the financial variables.

All analyses were performed using R version 3.4.0 [31].

## Results

### Data description and preliminary analyses

From 1889–1931, there were 86,023 typhoid deaths across all cities (median: 3,382 deaths per city). S3 Fig shows the weekly time series of typhoid mortality in each city. Of the 16 cities, four used reservoirs or lakes as their water source, three drew water from the Great Lakes, and nine accessed water from rivers (Table 2; additional details in S2 Table). Most cities introduced water chlorination or filtration during the study period, but some cities implemented other interventions. Boston's Metropolitan Water District completed a new reservoir in 1908, while New York built several additional reservoirs between 1905–1915. The Sanitary District of Chicago changed the direction of flow of the Chicago River so sewage from the city would no longer be discharged into Lake Michigan, the city's water source. To address flooding problems from periodic hurricanes and its location below sea level, the New Orleans Drainage Commission began to periodically drain the water supply in 1900. San Francisco had no water supply interventions that we could identify; however, a major earthquake in 1906 resulted in severe infrastructure damage and changes to the water supply system, and was included as a proxy intervention in our analysis.

In the preliminary harmonic regression analysis, fluctuations in typhoid mortality generally became less extreme from pre- to post-intervention periods. The six-month amplitude in typhoid mortality decreased in all but two cities (Milwaukee and Nashville), while the one-year seasonal amplitude decreased in all cities but New Orleans post-intervention (S4 and S5 Figs, S3 Table). In the two cities where the six-month amplitude increased, the amplitude was already extremely low in the pre-intervention period and did not increase by much in the post-intervention period. In every city, typhoid mortality significantly decreased with time in the post-intervention period. The pre-intervention time trend was less consistent across cities.

**Table 2. Descriptive statistics of cities and their water supplies.** "Total Deaths" are the number reported after imputation for missing data. Missing data numbers represent estimates after correcting for "true zeros" in the datasets, and before imputation.

| City | State | Total Deaths 1889–1931 | % (Number) Weekly Missing Mortality Data | Population in 1888 | Water Source Type | Year of (1st) Intervention | Type of Water Supply Intervention(s) 1889–1931 |
|---|---|---|---|---|---|---|---|
| Baltimore | MD | 5,198 | 4.5% (100) | 431,000 | Reservoirs | 1910 | Chlorination; Filtration |
| Boston | MA | 3,412 | 5.4% (117) | 414,000 | Lakes/ Reservoirs | 1908 | New reservoir |
| Chicago | IL | 13,161 | 6.8% (150) | 981,000 | Great Lake | 1900 | Changed river flow; Chlorination |
| Cincinnati | OH | 3,292 | 7.5% (167) | 289,000 | River | 1908 | Chlorination; Filtration |
| Cleveland | OH | 3,622 | 5.1% (115) | 241,000 | Great Lake | 1913 | Chlorination; Filtration |
| Milwaukee | WI | 1,912 | 16.0% (358) | 187,000 | Great Lake | 1910 | Chlorination |
| Nashville | TN | 1,535 | 10.2% (227) | 69,594 | River | 1908 | Chlorination; Filtration |
| New Orleans | LA | 3,352 | 2.0% (45) | 237,000 | River | 1900 | Drainage; Filtration |
| New York | NY | 16,991 | 3.5% (79) | 2,370,000 | Reservoirs | 1903 | New Reservoirs; Chlorination; Filtration |
| Philadelphia | PA | 13,927 | 16.3% (364) | 1,010,000 | River | 1902 | Chlorination; Filtration |
| Pittsburgh | PA | 7,864 | 17.3% (386) | 322,000 | River | 1908 | Chlorination; Filtration |
| Providence | RI | 1,106 | 13.1% (294) | 127,000 | River | 1902 | Filtration |
| Saint Louis | MO | 3,271 | 21.9% (490) | 432,000 | River | 1904 | Chlorination; Filtration |
| San Francisco | CA | 2,348 | 17.6% (393) | 286,000 | Lakes/ Reservoirs | 1906 | Earthquake* |
| Toledo | OH | 1,381 | 22.8% (510) | 75,167 | River | 1910 | Chlorination; Filtration |
| Washington | DC | 3,651 | 5.1% (113) | 214,000 | River | 1903 | Chlorination; Filtration |

*No interventions were identified for San Francisco, but the 1906 earthquake was used as a proxy due to the necessary infrastructure improvements that followed.

While the harmonic regression analyses suggested changes in the seasonality of typhoid mortality following interventions, there was little to no difference in seasonality of typhoid transmission pre- versus post-intervention estimated using TSIR models upon visual inspection (S6 Fig). Thus, we estimated the seasonal transmission rate for the entire 43-year study period in subsequent analyses. The similarity between pre- and post-intervention seasonality in the TSIR models but not in the harmonic regression models in the preliminary analyses suggests the need for using models that incorporate disease dynamics as opposed to simpler analyses that do not take disease dynamics into account (S4 and S6 Figs).

## Variations in seasonal patterns

Based on the full TSIR model (including waning of immunity), seasonal typhoid transmission increased at the beginning of the year and peaked around late summer or early fall in most cities (months 8–10; Fig 1, S4 Table). This trend varied somewhat across cities. In New Orleans,

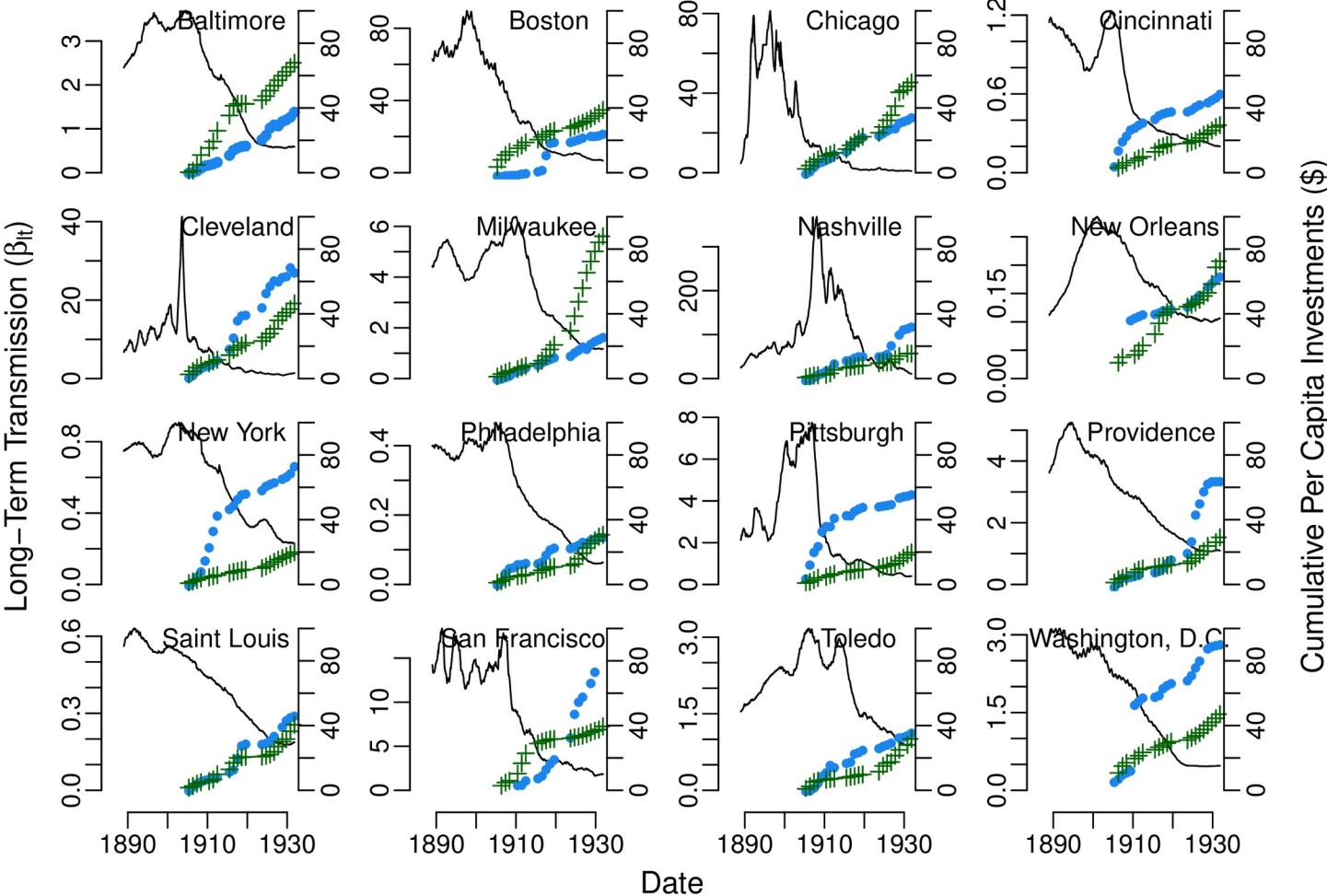

**Fig 1. Annual seasonal typhoid transmission estimated from Time-series Susceptible-Infectious-Recovered models.** The estimated seasonal transmission rate in each 4-week period is plotted for each city (color-coded by water source type; solid lines are the mean estimates and dashed lines are the 95% confidence intervals). The second-to-last panel shows the mean seasonal transmission across all cities in bold black. The last panel shows the mean seasonal transmission rate for cities with a particular water source type, with reservoirs in blue, rivers in green, and Great Lakes in purple. Seasons are shown in the background in shades of grey (medium-light grey for winter, light grey for spring, dark grey for summer, and medium-dark grey for fall).

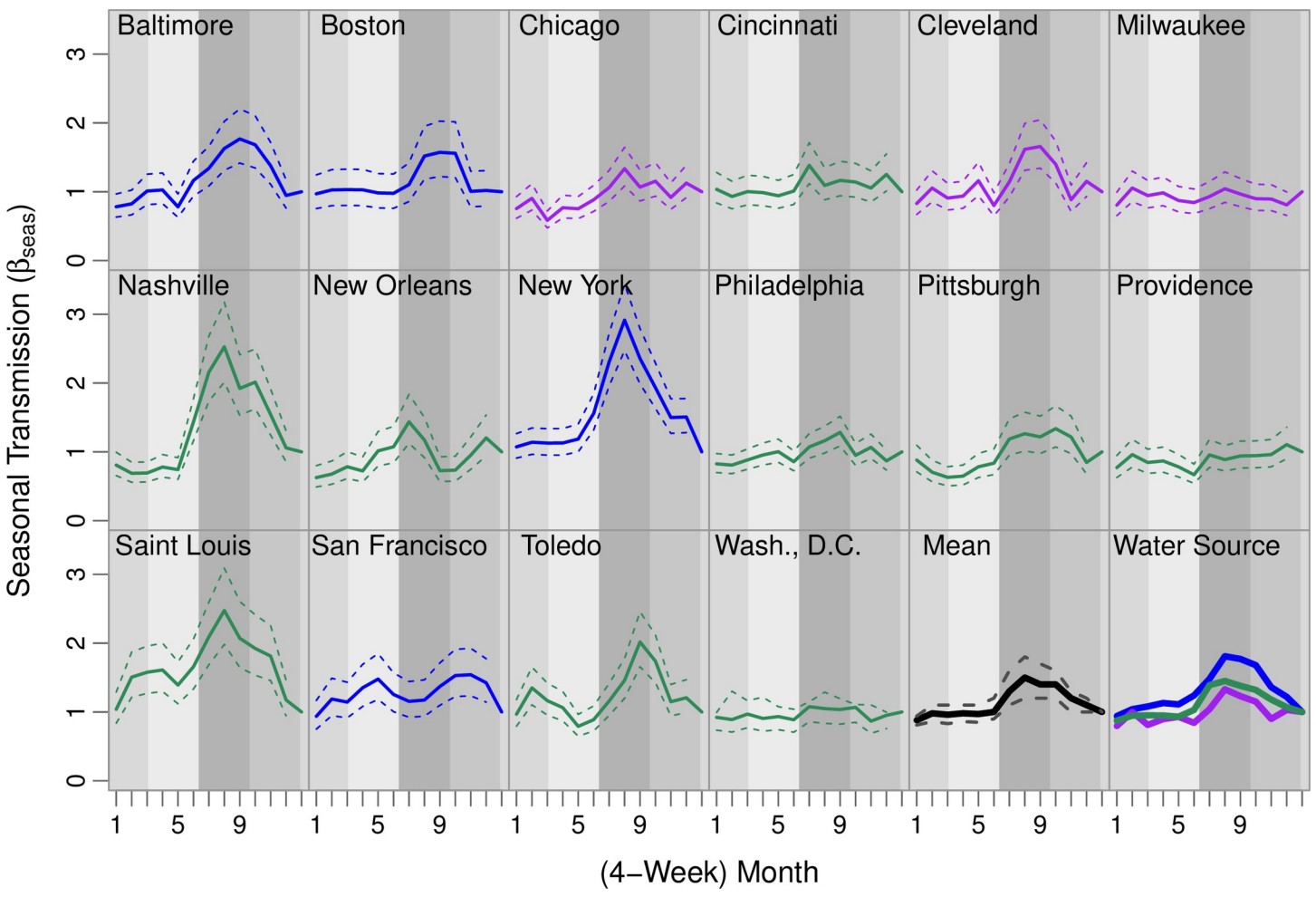

**Fig 2. Long-term typhoid transmission rate by city estimated from Time-series Susceptible-Infectious-Recovered models.** The estimated long-term transmission rate ($\beta_{lt}$, solid black line) is plotted for each city, by four-week generation interval. Overall per capita investments in the water supply (blue circles) and sewer system (green pluses) in 1931 US dollars are also shown for each city from 1902–1931.

peak transmission occurred earlier (months 7), while in San Francisco the peak occurred later (months 10–11). In several cities, there were additional peaks in the winter (months 1–3).

Seasonality in typhoid transmission also varied by water source type. While the seasonal trend was similar across different water source types, the magnitude of the peaks in transmission differed (bottom-right panel of Fig 1, S4 Table). Cities that relied on reservoirs had the highest amplitude of seasonal typhoid transmission, while cities that drew water from the Great Lakes had the least variability.

## Long-term typhoid transmission and investments in water and sanitation

After the 1900s, long-term typhoid transmission began to decrease almost monotonically in every city (Fig 2). Conversely, overall investments in water and sewer systems increased over time (Fig 2). Overall investments in both the water supply and sewer system were significantly associated with long-term typhoid transmission. Each $1 (in 1931) per capita increase in overall cumulative investment in water and sewer systems was associated with an estimated average 5% (95% confidence interval: 3–6%) and 6% (95% confidence interval: 4–9%) decrease in

**Table 3. Results of hierarchical regression analyses for overall investment variables: Random and fixed effects for yearly average long-term typhoid transmission vs. overall investments in water and sewer systems.** Each estimate shows the associated multiplicative change in the estimated long-term typhoid transmission rate for each $1 per capita increase in overall investment for the water supply and sewer system (in 1931 US dollars). Both random and fixed effects are shown, with their 95% confidence intervals.

| | | Estimate | |
|---|---|---|---|
| | | *Water Supply* | *Sewer System* |
| *Random + Fixed* | Baltimore | 0.95 (0.93–0.96) | 0.97 (0.95–0.99) |
| | Boston | 0.94 (0.93–0.96) | 0.93 (0.91–0.96) |
| | Chicago | 0.91 (0.90–0.93) | 0.95 (0.92–0.97) |
| | Cincinnati | 0.95 (0.94–0.97) | 0.95 (0.92–0.97) |
| | Cleveland | 0.97 (0.95–0.98) | 0.94 (0.92–0.97) |
| | Milwaukee | 0.93 (0.91–0.94) | 0.98 (0.96–1.00) |
| | Nashville | 0.91 (0.90–0.93) | 0.82 (0.79–0.85) |
| | New Orleans | 0.97 (0.95–0.99) | 0.98 (0.96–1.01) |
| | New York | 0.98 (0.97–1.00) | 0.93 (0.90–0.96) |
| | Philadelphia | 0.93 (0.91–0.95) | 0.93 (0.91–0.96) |
| | Pittsburgh | 0.94 (0.92–0.95) | 0.85 (0.82–0.88) |
| | Providence | 0.98 (0.97–1.00) | 0.95 (0.92–0.98) |
| | Saint Louis | 0.99 (0.98–1.00) | 0.94 (0.92–0.97) |
| | San Francisco | 0.98 (0.96–0.99) | 0.97 (0.94–0.99) |
| | Toledo | 0.96 (0.95–0.98) | 0.95 (0.93–0.98) |
| | Washington, D.C. | 0.98 (0.97–0.99) | 0.94 (0.92–0.97) |
| *Fixed* | - | 0.95 (0.94–0.97) | 0.94 (0.91–0.96) |

typhoid transmission, respectively (Table 3). Overall investments in both the water supply and sewer system were also significantly inversely associated (i.e. confidence interval entirely below one) with city-level transmission in 15 of the 16 cities (Table 3). The proportion of variability in long-term typhoid transmission explained by the both the fixed effects and random effects for overall investments was 98% for both variables, while average overall investments (i.e. fixed effects alone) explained 33% and 28% of the variability in typhoid transmission for the water supply and sewer system, respectively (S5 Table).

When considering the other financial variables, the associations were not as consistent across cities. Annual investments in maintenance or operation (receipts or expenses) had more city-level associations as compared to acquisition or construction variables (outlays) (S6 Table). In some instances, the relationship between the individual investment variables and typhoid transmission was positive (S6 Table, S7–S16 Figs).

## TSIR model fit

In general, the TSIR models fit to the full 43-year time series provided an adequate fit to the data. The full TSIR models (including waning of immunity) explained approximately 66% (range: 45–90%) of the variability in typhoid mortality counts over the study period (S7 Table). When we validated the models by fitting to the data through 1926 then using the fitted models to predict the last five years of typhoid mortality, in most cases the overall predicted trend and seasonal peaks in typhoid mortality were captured, but the model could not explain some of the mortality spikes (S17–S20 Figs). Nevertheless, the models generally provided a good fit to the data, with small out-of-sample mean squared prediction errors (S5 Table).

Our results were not sensitive to methods of handling missing data and zeros or variations in model structure (S2 Text, S8–S10 Tables). Seasonal transmission patterns remained the

same, and long-term trends retained their general shape (S2 Text, S21–S36 Figs). Our results were also not sensitive to the threshold for the maximum duration of immunity (S8 and S9 Tables). All cities had different patterns and functions of immunity decay, but the shapes of the seasonal and long-term transmission rates of typhoid were mostly preserved when the models were fit assuming the maximum duration of immunity (173 generation intervals, or approximately 13 years) or no waning of immunity.

## Discussion

The decline in typhoid mortality in the early 20th century U.S. has been attributed to investments in water and sewer systems. Our analysis strengthens this hypothesis. Furthermore, we characterized seasonal and long-terms trends in typhoid transmission and quantified the relationship between overall infrastructure investments and declines in transmission rates.

Historically, typhoid fever cases peaked during late summer/early fall in the U.S. [6, 32]. Yearly peaks of typhoid transmission coincide with warmer temperatures, similar to global trends [7, 33–35]. This pattern may be related to the enhanced growth of the bacteria at warmer temperatures, seasonal changes in diet (i.e. increased consumption of uncooked fruit and vegetables in summer and fall), or the increased abundance of flies that may serve as mechanical vectors of the bacteria [6, 7, 36, 37]. Additional fluctuations in typhoid transmission seen in some cities might be explained by seasonal variation in rainfall, which typically peaks in spring and summer in the eastern U.S. and winter on the west coast, and can impact the water supply to a city [34, 35, 38].

The overall amplitudes in typhoid seasonality did not appear to cluster geographically (S5 Fig), which led us to the hypothesis that the differences between cities may be due to differences in water source type. Variations due to water source type have a number of possible explanations. Cities relying on the Great Lakes for water had the least seasonal variability in transmission. Large bodies of water tend to be less impacted by seasonal changes in temperature and rainfall [39, 40]. The Great Lakes have a moderating effect on climate, absorbing heat and cooling the air in the summer, yet radiating heat and protecting from frost in the fall [41, 42]. Flowing water can slow down the movement of microbes [43], which may explain the lower seasonal variability in typhoid transmission among cities that draw water from rivers. Reservoirs and lakes are mostly smaller stagnant water sources, and may be more sensitive to seasonal changes in climate.

Differences in water source type may help to explain why some nearby cities exhibited different seasonal patterns. For example, New York and Philadelphia, though less than 100 miles apart, had different patterns of seasonal typhoid mortality and transmission (Fig 1, S3–S5 Figs). From 1890–1910, the typhoid mortality rate in New York was considerably lower than in Philadelphia (22.4 versus 43.1 deaths per 100,000 people per year, respectively). However, typhoid transmission was more seasonal in New York (which relied on rural reservoirs) compared to Philadelphia (which drew its water from rivers running through the city). While typhoid transmission consistently peaked in the late summer/early fall in New York, Philadelphia had only small seasonal variations in the transmission rate. It is possible that these differences reflect differences in the predominant route of typhoid transmission (i.e. food- versus water-borne) in the two cities. Strong seasonality in typhoid incidence was also noted in Santiago, Chile in the 1970-80s, and was linked to seasonal irrigation of crops with contaminated wastewater; typhoid incidence declined sharply once this practice was ended [5, 44, 45]. A better understanding of the drivers underlying seasonal patterns of typhoid transmission, and the differences noted among the various water sources, can aid typhoid control efforts.

Overall investments in the water supply and sewer system were inversely associated with long-term typhoid transmission in every city. These two predictors explained most of the

variability in long-term typhoid transmission when taking into account city-level random effects. These findings demonstrate the strong influence of investments in water and sanitation on typhoid transmission over time. However, other factors may also contribute. Associations also varied across cities, perhaps reflecting differences in water source types, public versus private ownership of water supplies, and rates of migration and poverty in the different cities.

A previous study by Cutler and Miller had similar findings [12]. They estimated that on average, filtration and chlorination reduced typhoid fever mortality by 25% from 1900 to 1936. They claimed that clean water technologies explained almost all of the decline in typhoid mortality, estimating that the cost of clean water technologies per person-year saved was $500 in 2003 ($666 in 2017), suggesting it was highly cost-effective. However, their analysis did not consider the complexities of typhoid transmission, such as chronic carriers, host immunity, and interactions between susceptible and infectious individuals, which makes it difficult to extrapolate their findings to better understand the impact of water and sanitation investments on typhoid transmission in modern contexts.

In the early 20<sup>th</sup> century, William Sedgwick studied what he referred to as the "Mills-Reincke Phenomenon", in which the introduction of sanitation and subsequent decrease in typhoid deaths was also associated with decreases in mortality from other diseases [46]. In the first half of the 20<sup>th</sup> century, all-cause mortality fell by 40% [12]. Typhoid fever and other waterborne diseases were not the only diseases to decline during this period; many non-enteric diseases were also reduced by 1931 [8].

It has thus far been difficult to evaluate the benefits of water and sanitation infrastructure investments compared to the deployment of new typhoid conjugate vaccines without data to quantify the costs and impact of the former [47]. With the recent World Health Organization recommendation for typhoid conjugate vaccine use and pilot studies underway [48, 49], governments are looking to prioritize the allocation of resources to yield the greatest decrease in typhoid burden. While long-term investments in water and sanitation systems are associated with decreased typhoid transmission, they also have benefits that extend beyond typhoid. Nevertheless, future studies should focus on comparing the cost-effectiveness and budget impact of the two interventions, bearing in mind the context and feasibility of deployment.

This study had some limitations. The weekly mortality counts likely suffer from lack of sensitivity and specificity in the diagnosis of typhoid fever. Additionally, we implicitly account for case fatality rates in our analysis. These issues are unlikely to bias our results provided the under- or over-reporting of typhoid mortality (and case fatality rate) was consistent over the study period. The cities chosen for our analysis were also limited by data availability. As a result, all of the cities were primarily in the northeastern U.S. All cities also had missing data, which had to be imputed. Furthermore, the roles of chronic carriers and immunity to typhoid are not fully understood. Our inclusion of carriers in the model matches the natural history of typhoid, but we did not examine whether it was necessary to model carriers separately. Patterns in the decay of immunity to typhoid varied widely across cities. Nevertheless, transmission rate estimates were not sensitive to the way we modelled immunity to infection. Finally, due to high levels of correlation between the financial variables, we were not able to estimate the combined effect of water supply and sewer system variables. Some of the overall decline in transmission may have been attributable to other interventions such as economic and nutritional gains, and behavior-change campaigns targeting hand and food washing [12, 50–52].

Our results aid in the understanding of the dynamics of typhoid transmission and potential impact of improvements in water and sanitation infrastructure, which is still lacking in many parts of the world. Before improvements in water and sanitation systems in the U.S., typhoid fever and other water-borne diseases were common. In 1900, infectious diseases (and typhoid in particular) accounted for 44% (2.4%) of deaths in major cities in the U.S. [8], compared to

30–51% (0.3–0.7%) in current day low- and middle-income countries [53–55]. Worldwide, approximately 1.1 billion people lack access to clean water, and roughly 2.5 billion people lack adequate sanitation [56]. Water and sanitation technologies can have substantial health returns; however, the continued operation and maintenance of these systems can be costly. Resource-poor countries must prioritize spending on public health issues, bearing in mind the cost-effectiveness and affordability of implementing and maintaining interventions. Our results can help to inform comparative cost-effectiveness analyses of different interventions to reduce the global burden of typhoid fever.

## Supporting information

**S1 Text. Model-fitting process.**
(DOCX)

**S2 Text. Sensitivity analyses.**
(DOCX)

**S1 Fig. Map of 16 cities with water supply types.** Each city included in the analysis is denoted by a different color in its geographical location in the United States. Squares denote cities with reservoirs, triangles denote those using the Great Lakes, and circles denote those with rivers as their main water source. The underlying map is adapted from the United States Geological Survey LandsatLook < https://landlook.usgs.gov/viewer.html# >.
(TIF)

**S2 Fig. Yearly reported population, extrapolated monthly population, and estimated susceptible population over study period.** The yearly U.S. Census Bureau reported population (red Xs), monthly population extrapolated using cubic splines (solid black line), and susceptible population (dashed black line) estimated from the main TSIR models are shown for each city over the study period. Note that in some cities, the susceptible and total population are very close and cannot be differentiated in the plots.
(TIF)

**S3 Fig. Weekly time-series of reported typhoid mortality in each city.** The observed (including imputation, in blue) time series of weekly deaths reportedly due to typhoid (black lines) and the yearly typhoid deaths per 100,000 people (red Xs) is shown for each city from 1889–1931.
(TIF)

**S4 Fig. Pre- -and post-intervention sinusoid curves from preliminary harmonic regression analyses.** The pre- (blue) and post-intervention (red) six- and 12-month sinusoid curves fitted to the typhoid mortality data are shown for each city, along with the seasonal transmission rate estimated by the main TSIR model (dashed black line).
(TIF)

**S5 Fig. Map of 12- and 6-month amplitudes of typhoid mortality counts, from preliminary harmonic regression analyses.** The average 12- and 6-month amplitudes of seasonal variation in reported typhoid mortality estimated from the harmonic regression analyses are shown separately according to the color scale and plotted by geographic location.
(TIF)

**S6 Fig. Seasonal transmission rate for pre- and post- water supply intervention periods.** The estimated four-week seasonal transmission rates extracted from each city's simple TSIR model (not including waning of immunity) are shown for each pre- (blue) and post- (red)

water supply intervention period.
(TIF)

**S7 Fig. Annual per capita water supply receipts.** Annual water supply receipts from 1902–1931 are shown for each city in per capita increments (US$ per person). Dollar amounts are adjusted for inflation to 1931 US$.
(TIF)

**S8 Fig. Annual per capita water supply expenses.** Annual spending on water supply expenses from 1902–1931 is shown for each city in per capita increments (US$ per person). Dollar amounts are adjusted for inflation to 1931 US$.
(TIF)

**S9 Fig. Annual per capita sewer system expenses.** Annual spending on sewer system expenses from 1902–1931 is shown for each city in per capita increments (US$ per person). Dollar amounts are adjusted for inflation to 1931 US$.
(TIF)

**S10 Fig. Annual per capita water supply outlays.** Annual spending on water supply outlays from 1902–1931 is shown for each city (green dots) in per capita increments (US$ per person). The year in which interventions were introduced are represented by the dashed lines for filtration (red), chlorination (blue), or other interventions (purple). The inclusion of intervention dates is for illustrative purposes. Outliers not seen: In 1930, water supply outlays from San Francisco totalled $70.97 per capita; this was the year in which the city purchased the water supply previously owned and operated by the Spring Valley Water Company. Chicago and New Orleans introduced water supply interventions in 1900, prior to the time period shown. Dollar amounts are adjusted for inflation to 1931 US$.
(TIF)

**S11 Fig. Annual per capita sewer system outlays.** Annual spending on sewer system outlays from 1902–1931 is shown for each city in per capita increments (US$ per person). Dollar amounts are adjusted for inflation to 1931 US$.
(TIF)

**S12 Fig. Annual per capita value of the water supply system.** The overall annual value of the water supply system from 1902–1931 is shown for each city in per capita increments (US$ per person). Outliers not seen: In 1897, the value of the water supply system totalled $508.55 per capita in Washington, D.C. Dollar amounts are adjusted for inflation to 1931 US$.
(TIF)

**S13 Fig. Annual per capita funded debt of the water supply system.** The overall annual accrued debt and/or funded loans for the water supply system from 1902–1931 is shown for each city in per capita increments (US$ per person). Note: Data were not available for this variable in Washington, D.C. Dollar amounts are adjusted for inflation to 1931 US$.
(TIF)

**S14 Fig. Annual per capita funded debt of the sewer system.** The overall annual accrued debt and/or funded loans for the sewer system from 1902–1931 are shown for each city in per capita increments (US$ per person). Note: Data were not available for this variable in Washington, D.C. Dollar amounts are adjusted for inflation to 1931 US$.
(TIF)

**S15 Fig. Overall investment in the water supply system.** The overall cumulative investments in the water supply system from 1902–1931 are shown for each city in per capita increments (US$ per person). This was calculated as the cumulative sum of annual expenses and annual outlays minus annual receipts for the water supply system. Dollar amounts are adjusted for inflation to 1931 US$.
(TIF)

**S16 Fig. Overall investment in the sewer system.** The overall cumulative investments in the sewer system from 1902–1931 are shown for each city in per capita increments (US$ per person). This was calculated as the cumulative sum of annual expenses and annual outlays for the sewer system. Dollar amounts are adjusted for inflation to 1931 US$.
(TIF)

**S17 Fig. TSIR model predictions for Baltimore, Boston, Chicago, and Cincinnati.** For each city, the TSIR model is fit using the first 38 years of data, then used to predict the last 5 years of data. In each plot, the observed data is shown in black, the model fit to the first 38 years is shown in blue, and the predicted last 5 years is shown in red.
(TIF)

**S18 Fig. TSIR model predictions for Cleveland, Milwaukee, Nashville, and New Orleans.** For each city, the TSIR model is fit using the first 38 years of data, then used to predict the last 5 years of data. In each plot, the observed data is shown in black, the model fit to the first 38 years is shown in blue, and the predicted last 5 years is shown in red.
(TIF)

**S19 Fig. TSIR model predictions for New York, Philadelphia, Pittsburgh, and Providence.** For each city, the TSIR model is fit using the first 38 years of data, then used to predict the last 5 years of data. In each plot, the observed data is shown in black, the model fit to the first 38 years is shown in blue, and the predicted last 5 years is shown in red.
(TIF)

**S20 Fig. TSIR model predictions for St. Louis, San Francisco, Toledo, and Washington, D. C.** For each city, the TSIR model is fit using the first 38 years of data, then used to predict the last 5 years of data. In each plot, the observed data is shown in black, the model fit to the first 38 years is shown in blue, and the predicted last 5 years is shown in red.
(TIF)

**S21 Fig. Plots of seasonal transmission from sensitivity analyses for imputation, log-transformation, and duration of immunity: Baltimore.** The plot of seasonal and long-term transmission is shown for each city separately. Plots are shown for imputation of missing data (8-, 13-, and 26-week algorithm); addition to all weekly death counts (+1 shown in the solid line in every panel; +0.5 shown in the dashed line in the second panel); and duration of immunity (13-year, 1-year, and no waning of immunity) in the Baltimore data.
(TIF)

**S22 Fig. Plots of seasonal transmission from sensitivity analyses for imputation, log-transformation, and duration of immunity: Boston.** The plot of seasonal and long-term transmission is shown for each city separately. Plots are shown for imputation of missing data (8-, 13-, and 26-week algorithm); addition to all weekly death counts (+1 shown in the solid line in every panel; +0.5 shown in the dashed line in the second panel); and duration of immunity (13-year, 1-year, and no waning of immunity) in the Boston data.
(TIF)

**S23 Fig. Plots of seasonal transmission from sensitivity analyses for imputation, log-transformation, and duration of immunity: Chicago.** The plot of seasonal and long-term transmission is shown for each city separately. Plots are shown for imputation of missing data (8-, 13-, and 26-week algorithm); addition to all weekly death counts (+1 shown in the solid line in every panel; +0.5 shown in the dashed line in the second panel); and duration of immunity (13-year, 1-year, and no waning of immunity) in the Chicago data.
(TIF)

**S24 Fig. Plots of seasonal transmission from sensitivity analyses for imputation, log-transformation, and duration of immunity: Cincinnati.** The plot of seasonal and long-term transmission is shown for each city separately. Plots are shown for imputation of missing data (8-, 13-, and 26-week algorithm); addition to all weekly death counts (+1 shown in the solid line in every panel; +0.5 shown in the dashed line in the second panel); and duration of immunity (13-year, 1-year, and no waning of immunity) in the Cincinnati data.
(TIF)

**S25 Fig. Plots of seasonal transmission from sensitivity analyses for imputation, log-transformation, and duration of immunity: Cleveland.** The plot of seasonal and long-term transmission is shown for each city separately. Plots are shown for imputation of missing data (8-, 13-, and 26-week algorithm); addition to all weekly death counts (+1 shown in the solid line in every panel; +0.5 shown in the dashed line in the second panel); and duration of immunity (13-year, 1-year, and no waning of immunity) in the Cleveland data. Note that the 26-week imputation plot is not shown entirely in the plot due to its outlier.
(TIF)

**S26 Fig. Plots of seasonal transmission from sensitivity analyses for imputation, log-transformation, and duration of immunity: Milwaukee.** The plot of seasonal and long-term transmission is shown for each city separately. Plots are shown for imputation of missing data (8-, 13-, and 26-week algorithm); addition to all weekly death counts (+1 shown in the solid line in every panel; +0.5 shown in the dashed line in the second panel); and duration of immunity (13-year, 1-year, and no waning of immunity) in the Milwaukee data.
(TIF)

**S27 Fig. Plots of seasonal transmission from sensitivity analyses for imputation, log-transformation, and duration of immunity: Nashville.** The plot of seasonal and long-term transmission is shown for each city separately. Plots are shown for imputation of missing data (8-, 13-, and 26-week algorithm); addition to all weekly death counts (+1 shown in the solid line in every panel; +0.5 shown in the dashed line in the second panel); and duration of immunity (13-year, 1-year, and no waning of immunity) in the Nashville data.
(TIF)

**S28 Fig. Plots of seasonal transmission from sensitivity analyses for imputation, log-transformation, and duration of immunity: New Orleans.** The plot of seasonal and long-term transmission is shown for each city separately. Plots are shown for imputation of missing data (8-, 13-, and 26-week algorithm); addition to all weekly death counts (+1 shown in the solid line in every panel; +0.5 shown in the dashed line in the second panel); and duration of immunity (13-year, 1-year, and no waning of immunity) in the New Orleans data.
(TIF)

**S29 Fig. Plots of seasonal transmission from sensitivity analyses for imputation, log-transformation, and duration of immunity: New York.** The plot of seasonal and long-term transmission is shown for each city separately. Plots are shown for imputation of missing data (8-,

13-, and 26-week algorithm); addition to all weekly death counts (+1 shown in the solid line in every panel; +0.5 shown in the dashed line in the second panel); and duration of immunity (13-year, 1-year, and no waning of immunity) in the New York data.
(TIF)

**S30 Fig. Plots of seasonal transmission from sensitivity analyses for imputation, log-transformation, and duration of immunity: Philadelphia.** The plot of seasonal and long-term transmission is shown for each city separately. Plots are shown for imputation of missing data (8-, 13-, and 26-week algorithm); addition to all weekly death counts (+1 shown in the solid line in every panel; +0.5 shown in the dashed line in the second panel); and duration of immunity (13-year, 1-year, and no waning of immunity) in the Philadelphia data.
(TIF)

**S31 Fig. Plots of seasonal transmission from sensitivity analyses for imputation, log-transformation, and duration of immunity: Pittsburgh.** The plot of seasonal and long-term transmission is shown for each city separately. Plots are shown for imputation of missing data (8-, 13-, and 26-week algorithm); addition to all weekly death counts (+1 shown in the solid line in every panel; +0.5 shown in the dashed line in the second panel); and duration of immunity (13-year, 1-year, and no waning of immunity) in the Pittsburgh data. Note that the imputed 13-week algorithm (+0.5) and the imputed 26-week algorithm (+1) are not shown entirely in the plots due to outliers.
(TIF)

**S32 Fig. Plots of seasonal transmission from sensitivity analyses for imputation, log-transformation, and duration of immunity: Providence.** The plot of seasonal and long-term transmission is shown for each city separately. Plots are shown for imputation of missing data (8-, 13-, and 26-week algorithm); addition to all weekly death counts (+1 shown in the solid line in every panel; +0.5 shown in the dashed line in the second panel); and duration of immunity (13-year, 1-year, and no waning of immunity) in the Providence data.
(TIF)

**S33 Fig. Plots of seasonal transmission from sensitivity analyses for imputation, log-transformation, and duration of immunity: Saint Louis.** The plot of seasonal and long-term transmission is shown for each city separately. Plots are shown for imputation of missing data (8-, 13-, and 26-week algorithm); addition to all weekly death counts (+1 shown in the solid line in every panel; +0.5 shown in the dashed line in the second panel); and duration of immunity (13-year, 1-year, and no waning of immunity) in the Saint Louis data.
(TIF)

**S34 Fig. Plots of seasonal transmission from sensitivity analyses for imputation, log-transformation, and duration of immunity: San Francisco.** The plot of seasonal and long-term transmission is shown for each city separately. Plots are shown for imputation of missing data (8-, 13-, and 26-week algorithm); addition to all weekly death counts (+1 shown in the solid line in every panel; +0.5 shown in the dashed line in the second panel); and duration of immunity (13-year, 1-year, and no waning of immunity) in the San Francisco data.
(TIF)

**S35 Fig. Plots of seasonal transmission from sensitivity analyses for imputation, log-transformation, and duration of immunity: Toledo.** The plot of seasonal and long-term transmission is shown for each city separately. Plots are shown for imputation of missing data (8-, 13-, and 26-week algorithm); addition to all weekly death counts (+1 shown in the solid line in every panel; +0.5 shown in the dashed line in the second panel); and duration of immunity

(13-year, 1-year, and no waning of immunity) in the Toledo data.
(TIF)

**S36 Fig. Plots of seasonal transmission from sensitivity analyses for imputation, log-transformation, and duration of immunity: Washington, D.C..** The plot of seasonal and long-term transmission is shown for each city separately. Plots are shown for imputation of missing data (8-, 13-, and 26-week algorithm); addition to all weekly death counts (+1 shown in the solid line in every panel; +0.5 shown in the dashed line in the second panel); and duration of immunity (13-year, 1-year, and no waning of immunity) in the Washington, D.C. data.
(TIF)

**S1 Table. References for water supply source, interventions and dates.** Information on water supply interventions and water sources were extracted from a variety of references, noted below. Most cities had data available from the U.S. Census Bureau in addition to individual municipal sources, noted in the table as "U.S. Census Bureau (Yes/No)".
(XLSX)

**S2 Table. Initial and estimated values for main TSIR models.** Initial parameters (median susceptible population, median overall population, infectious, susceptible, and newborn populations) and values estimated from the TSIR models (chronic carriers, underreporting factors, and heterogeneous mixing parameters) are shown for each city.
(XLSX)

**S3 Table. Harmonic regression analyses of typhoid mortality data pre- and post- water supply intervention.** Time trends and seasonal amplitudes were estimated for each city pre- and post- intervention in preliminary analyses with harmonic regression. Values shown in grey were not statistically significant at the 0.05-level, while values in black had p-values<0.05. In the last column, the ratio (post-/pre- water supply intervention) was calculated from the six-month and one-year amplitudes estimated from the regression models.
(XLSX)

**S4 Table. Estimates of seasonal transmission from TSIR models, with confidence intervals.** Results of the seasonal transmission parameters estimated from the TSIR models are shown. In the top half of the table, the estimated values for each four-week month's typhoid transmission compared to the 13th month are shown with their 95% confidence intervals, by city. Estimates with confidence intervals that are entirely below one are shown in red, and those with confidence intervals entirely above one are shown in blue. In the bottom half of the table, the percentage of each water type with confidence intervals entirely below or above one are shown for each month, highlighted from lighter to darker red or blue indicating the magnitude of the percentage.
(XLSX)

**S5 Table. Variability in long-term typhoid transmission explained by financial water supply and sewer system variables.** Values shown are the conditional and marginal $R^2$ from the hierarchical linear regression analyses for each financial variable.
(XLSX)

**S6 Table. Random and fixed effects for associations between yearly average long-term typhoid transmission and investments in water and sewer systems for individual financial variables.** Each estimate shows the associated change (and 95% confidence interval) in typhoid transmission for each $1 (1931 US$) per capita increase in the financial variable for the water supply (WS) and sewer system (SS) for fixed and random effects. No data were available for

Washington, D.C. for the variables Funded Debt of the Water Supply System or Funded Debt of the Sewer System.
(XLSX)

**S7 Table. Fit of the TSIR models to within- and out-of-sample data for each city.** Variability in typhoid deaths explained ($R^2$) by TSIR models fit to data for the full study period (1889–1931) is shown, along with the within-sample mean squared errors (MSE) for 1922–1926 (i.e. the last five years of within-sample data used to generate the prediction model), the out-of-sample mean squared prediction errors (MSPE) for 1927–1931 (i.e. the out-of-sample data), and their ratio (MSPE/MSE) for comparison.
(XLSX)

**S8 Table. Sensitivity analyses for hierarchical regression: Random and fixed effects for yearly average long-term typhoid transmission vs. overall investments in the water supply system.** Each estimate shows the associated multiplicative change in the estimated long-term typhoid transmission rate for each $1 per capita increase in overall investment for the water supply system (in 1931 US dollars) for each model fit. Both random and fixed effects are shown, with their 95% confidence intervals.
(XLSX)

**S9 Table. Sensitivity analyses for hierarchical regression: Random and fixed effects for yearly average long-term typhoid transmission vs. overall investments in the sewer system.** Each estimate shows the associated multiplicative change in the estimated long-term typhoid transmission rate for each $1 per capita increase in overall investment for the sewer system (in 1931 US dollars) for each model fit. Both random and fixed effects are shown, with their 95% confidence intervals.
(XLSX)

**S10 Table. Heterogeneous mixing from sensitivity analyses for assumptions of waning immunity, chronic carriers, or using a simple TSIR model.** Values are shown for each city and assumption. The second column shows the heterogeneous mixing parameter value in the final models fit, the "No K" column shows the value for models fit without including waning immunity, the "No C" column shows the value for models excluding chronic carriers as part of the transmission process, the "No K, No C" column shows the values for models excluding both waning immunity and chronic carrier populations, and the last column ("Simple TSIR Model") shows the value for models fit using ordinary least squares regression and does not use splines or smoothing weights.
(XLSX)

## Author Contributions

**Conceptualization:** Bryan T. Grenfell, Virginia E. Pitzer.

**Data curation:** Maile T. Phillips, Katharine A. Owers, Virginia E. Pitzer.

**Formal analysis:** Maile T. Phillips.

**Funding acquisition:** Bryan T. Grenfell, Virginia E. Pitzer.

**Investigation:** Maile T. Phillips.

**Methodology:** Maile T. Phillips, Katharine A. Owers, Bryan T. Grenfell, Virginia E. Pitzer.

**Software:** Maile T. Phillips.

**Validation:** Maile T. Phillips.

**Visualization:** Maile T. Phillips.

**Writing – original draft:** Maile T. Phillips, Virginia E. Pitzer.

**Writing – review & editing:** Maile T. Phillips, Katharine A. Owers, Bryan T. Grenfell, Virginia E. Pitzer.

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
