## [Decision Letter · Decision Letter 0]

25 Jul 2019

Dear Mrs. Phillips:

Thank you very much for submitting your manuscript "Changes in historical typhoid transmission across 16 U.S. cities, 1889-1931: Quantifying the impact of investments in water and sewer infrastructures" (#PNTD-D-19-00963) for review by PLOS Neglected Tropical Diseases. Your manuscript was fully evaluated at the editorial level and by independent peer reviewers. The reviewers appreciated the attention to an important problem, but raised some substantial concerns about the manuscript as it currently stands. These issues must be addressed before we would be willing to consider a revised version of your study. We cannot, of course, promise publication at that time.

We therefore ask you to modify the manuscript according to the review recommendations before we can consider your manuscript for acceptance. Your revisions should address the specific points made by each reviewer. 

When you are ready to resubmit, please be prepared to upload the following:

(1) A letter containing a detailed list of your responses to the review comments and a description of the changes you have made in the manuscript.

(2) Two versions of the manuscript: one with either highlights or tracked changes denoting where the text has been changed (uploaded as a "Revised Article with Changes Highlighted" file); the other a clean version (uploaded as the article file).

(3) If available, a striking still image (a new image if one is available or an existing one from within your manuscript). If your manuscript is accepted for publication, this image may be featured on our website. Images should ideally be high resolution, eye-catching, single panel images; where one is available, please use 'add file' at the time of resubmission and select 'striking image' as the file type. 

Please provide a short caption, including credits, uploaded as a separate "Other" file. If your image is from someone other than yourself, please ensure that the artist has read and agreed to the terms and conditions of the Creative Commons Attribution License at http://journals.plos.org/plosntds/s/content-license (NOTE: we cannot publish copyrighted images). 

(4) If applicable, we encourage you to add a list of accession numbers/ID numbers for genes and proteins mentioned in the text (these should be listed as a paragraph at the end of the manuscript). You can supply accession numbers for any database, so long as the database is publicly accessible and stable. Examples include LocusLink and SwissProt.

(5) To enhance the reproducibility of your results, we recommend that you deposit your laboratory protocols in protocols.io, where a protocol can be assigned its own identifier (DOI) such that it can be cited independently in the future. For instructions see http://journals.plos.org/plosntds/s/submission-guidelines#loc-methods

While revising your submission, please upload your figure files to the Preflight Analysis and Conversion Engine (PACE) digital diagnostic tool, https://pacev2.apexcovantage.com/ PACE helps ensure that figures meet PLOS requirements. To use PACE, you must first register as a user. Then, login and navigate to the UPLOAD tab, where you will find detailed instructions on how to use the tool. If you encounter any issues or have any questions when using PACE, please email us at figures@plos.org.

We hope to receive your revised manuscript by Sep 23 2019 11:59PM. If you anticipate any delay in its return, we ask that you let us know the expected resubmission date by replying to this email.

To submit a revision, go to https://www.editorialmanager.com/pntd/ and log in as an Author. You will see a menu item call Submission Needing Revision. You will find your submission record there. 

Sincerely,

Andrew S. Azman

Deputy Editor

Reviewer's Responses to Questions

**Key Review Criteria Required for Acceptance?**

**Methods**

-Are the objectives of the study clearly articulated with a clear testable hypothesis stated?

-Is the study design appropriate to address the stated objectives?

-Is the population clearly described and appropriate for the hypothesis being tested?

-Is the sample size sufficient to ensure adequate power to address the hypothesis being tested?

-Were correct statistical analysis used to support conclusions?

-Are there concerns about ethical or regulatory requirements being met?

Reviewer #1: I have two high-level questions about the methods:

 1. I don't know why long-term beta is log-transformed and the financial variables are linear in the regressions (page 10, lines 194-195). Were other transformations tested?

 2. There are many financial variables, and I found it difficult to track all the univariate associations with typhoid (lines 270-302). Would a multivariate analysis be feasible? Or dimensionality reduction? Or some correction for multiple hypothesis testing? Is there a way to convey how correlated these variables are, since they are generally monotonically increasing and hard to tell apart without effort (I had to look at several supporting figures to see this for myself)?

Reviewer #2: - Objectives of the study are clearly articulated.

- However I have some concerns, regarding the statistical method used to analyze time-series (see my general comments)

Reviewer #3: (No Response)

**Results**

-Does the analysis presented match the analysis plan?

-Are the results clearly and completely presented?

-Are the figures (Tables, Images) of sufficient quality for clarity?

Reviewer #1: The results were generally clear, though they are complex and required a second reading for me. Here are some issues I had:

 1. What method was used by imputeTS (page 8, line 137)?

 2. Are there any constraints on the two betas (seasonal and long-term)? I thought the seasonal beta values might have a floor or mean of 1.0 to reduce the degrees of freedom a bit, but that does not seem to be the case. In a related issue, in Figure 1 (page 13) I assume that a value of "1" is not special (e.g., a baseline value). The lowest values appear to be close to 1, so that's why I'm wondering how these values are anchored or maybe normalized. Understanding how beta_seas is constrained would make it easier to interpret beta_lt in Figure 2. For example, if beta_seas could not be below 1, then we could assume that there is net transmission (beta>1) when beta_lt is greater than 1. 

 3. In Figure S3, the y-axis for beta_seas goes up to 10, while it only reaches 3 in Figure 1. Why do the betas have similar shapes but much larger maximums in Figure 10?

 4. I don't know how to interpret the 6-month vs 12-month amplitudes from the harmonic regression (page 12, line 221). I was expecting to see a single seasonal peak decrease after interventions, but I don't know what to think about the 6-month amplitude if there is also a substantial 12-month amplitude that could also be out of phase. Should we think of this as a "minor" annual peak or something more complex? Would it be useful to plot the sinusoids identified in the regression so they could be compared with Figures 1 and S3?

 5. Where are the parameter values for the TSIR models, such as birth/death rates, alpha, C (carriers), duration of immunity, initial conditions? It would also be nice to know how much susceptibles are depleted each year, since this would help justify the use of a dynamic transmission model.

Reviewer #2: Yes, the analysis presented match the analysis plan.

Yes, the results are clearly and completely presented.

Yes, the figures are of sufficient quality.

Reviewer #3: (No Response)

**Conclusions**

-Are the conclusions supported by the data presented?

-Are the limitations of analysis clearly described?

-Do the authors discuss how these data can be helpful to advance our understanding of the topic under study?

-Is public health relevance addressed?

Reviewer #1: I think that the observation that the beta_seas values seemed pretty similar before and after intervention (page 12) might be one of the justifications of using a TSIR model instead of a simpler analysis with no disease dynamics (like the Cutler and Miller work mentioned on page 5). I wonder if this can be emphasized in the Discussion.

Do you think the Great Depression had any effect on the financial investments or typhoid reporting?

The investments in water and sanitation should have impacted most enteric diseases. Have there been relevant observations in the literature? Or hints of these trends in the Tycho database? Conversely, did non-enteric diseases not decline during this period?

Reviewer #2: - I fear the authors over-interpreted some of their results, notably because they neither took into account temporal auto-correlation in their time-series analysis, nor potential covariates. They should either try to improve the robustness of their analyses, or tone down their conclusions.

- Limitations of analysis are clearly described but their did not address the important question of temporal auto-correlation of their time series

- Authors could try to better link their historical results on typhoid in the US to the current situation in resource-poor countries. Onlly a couple of sentences deal with this in the discussion. It seems to me that the fact that investments in the continued operation and maintenance of water and sewer systems in the US had a larger and more immediate impact on typhoid transmission compared to investments in acquisition or construction could inspire important recommendations for development plans in resource-poor countries.

- The public health relevance of presented results could better be addressed.

Reviewer #3: (No Response)

**Editorial and Data Presentation Modifications?**

Reviewer #1: Minor issues and comments:

 1. Can you make a comment about the quality of typhoid mortality data from this time period (page 6, line 105). In the Discussion, the unknown sensitivity and specificity is mentioned, but maybe a qualitative statement about how typhoid deaths could have been ascertained would be nice.

 2. I can't tell if there are five vs six categories of financial data (page 7, lines 122 and 128).

 3. In Figure 1 (page 13), there should be more values printed on the x-axis.

 4. In Figure 2, what are the units for the investments (I assume cumulative $ per capita)? And would it be easier to read if the water and sewer symbols were different colors?

 5. In Figure S1, I can see only dots that represent cities. Should there be state or national boundaries or other context?

 6. In Figure S2, I find it really hard to see the number of weekly deaths. I think making the black line thinner could help. Would it be possible to indicate where imputation took place (maybe a different color)?

Reviewer #2: - Suggestion of short title : add « water supply » ?

- line 57 : add « in resource-poor countries » ?

- References : some reference don’t have the correct format and some URL don’t work

- line 83: is ref [6] proper here ?

- Ref [20] : use the correct reference of the package provided on CRAN: Moritz S, Bartz-Beielstein T (2017). “imputeTS: Time Series Missing Value Imputation in R.” The R Journal, 9(1), 207–218. doi: 10.32614/RJ-2017-009.

Reviewer #3: (No Response)

**Summary and General Comments**

Reviewer #1: The authors present an interesting overview and analysis of typhoid mortality trends and the investment in water and sanitation in 16 US cities at the beginning of the 20th century. While the general results seem reasonable, there are many covariates and outputs that are not necessarily intuitive, such as the numerous financial variables and the TSIR parameters, so it can be hard to understand the significance of the individual results. However, assembling these data and the sharing of the analysis code is a great contribution to the field and would allow other researchers to test alternative models. 

I made several suggestions that may require a lot of work to address, but I would be satisfied with short comments that address the concerns. For example, it is not clear to me if a multivariate analysis using all the financial variables would be reasonable, or if the authors could produce a simple plot or metric that shows how correlated the variables are.

Reviewer #2: The study submitted by MT Phillips and colleagues addresses a very interesting topic. 

Using historical databases of typhoid-associated deaths and financial investments in water and sewer systems from 1889-1931 in several 16 US cities, authors first fit transmission models. Extracting a seasonal component, they subsequently studied the seasonality of typhoid according to water source. Finally, they performed regression analyses between a long-term component of their models and financial variables. 

Authors showed that typhoid seasonality varied by water source, and calculated the respective impact of investments in water supply and sewer systems on typhoid transmission. Notably, authors found that : (1) investments in the water supply and sewer system were inversely associated with long-term typhoid transmission in every city ; (2) investments in sewer systems were not as strongly associated with trends in typhoid transmission as water supply variables ; and (3) investments in the continued operation and maintenance of water and sewer systems had a larger and more immediate impact on typhoid transmission compared to investments in acquisition or construction.

Their results may be valuable for typhoid control in resource-poor countries where the diseases keeps a high incidence.

In addition, putting together infectious disease transmission modeling and cost-impact analysis in water and sanitation is too rare and thus valuable. 

Nevertheless, their analyses raise several questions :

1/ I am not sure that authors used a proper method to analyze such time series, especially when associating typhoid long-term trend and WASH investments. Both phenomenons were concurrent or concomitant, and a simple linear regression analysis is thus not a recommended method here because it does not take into account temporal auto-correlation. Authors thus certainly over-interpreted their statistical results concerning this association, and I am not totally confident with their claim that financial predictors explained on average 46% of the variability in long-term typhoid transmission across the variables and cities.

Several alternative statistical approaches may be used: 

- Box-Jenkins methods with stationary time series

- Interrupted time series analysis 

- generalized additive models (GAM) with a spline function of time

2/ Authors acknowledged that additional factors than the source of water may explain the seasonality (rainfall, temperature…), and that additional factors than investments may explain the long-term trend of typhoid transmission (economic and nutritional gains, behavior-change on hand and food washing, health care…). This would be a great additional value to their study to include such factors in multivariate analyses. Several statistical approaches may be used : 

- SARIMAX

- generalized additive models (GAM) with a spline function of time

3/ I wonder what is the additional value of fitting Susceptible-Infectious-Recovered models compared to usual statistical models which assume less hypotheses and may be best suited to time-series analysis.

I recommend that authors at least discuss these elements and tone down their interpretation of presented results.

4/ Finally, authors could better link their historical results to current chalenges in typhoid and other water-borne diseases control in resource-poor countries.

Reviewer #3: (No Response)

PLOS authors have the option to publish the peer review history of their article (what does this mean?). If published, this will include your full peer review and any attached files.

Reviewer #1: No

Reviewer #2: Yes: Stanislas Rebaudet

Reviewer #3: Yes: Elizabeth Lee

---

## [Decision Letter · Decision Letter 1]

14 Nov 2019

Dear Mrs. Phillips:

Thank you very much for submitting your manuscript "Changes in historical typhoid transmission across 16 U.S. cities, 1889-1931: Quantifying the impact of investments in water and sewer infrastructures" (PNTD-D-19-00963R1) for review by PLOS Neglected Tropical Diseases. Your manuscript was fully evaluated at the editorial level and by independent peer reviewers. The reviewers appreciated the attention to an important topic but identified some aspects of the manuscript that should be improved.

We therefore ask you to modify the manuscript according to the review recommendations before we can consider your manuscript for acceptance. Your revisions should address the specific points made by each reviewer.

(1) A letter containing a detailed list of your responses to the review comments and a description of the changes you have made in the manuscript.

(2) Two versions of the manuscript: one with either highlights or tracked changes denoting where the text has been changed (uploaded as a "Revised Article with Changes Highlighted" file ); the other a clean version (uploaded as the article file).

(3) If available, a striking still image (a new image if one is available or an existing one from within your manuscript). If your manuscript is accepted for publication, this image may be featured on our website. Images should ideally be high resolution, eye-catching, single panel images; where one is available, please use 'add file' at the time of resubmission and select 'striking image' as the file type. 

Please provide a short caption, including credits, uploaded as a separate "Other" file. If your image is from someone other than yourself, please ensure that the artist has read and agreed to the terms and conditions of the Creative Commons Attribution License at http://journals.plos.org/plosntds/s/content-license (NOTE: we cannot publish copyrighted images). 

(4) Appropriate Figure Files 

Please remove all name and figure # text from your figure files upon submitting your revision. Please also take this time to check that your figures are of high resolution, which will improve both the editorial review process and help expedite your manuscript's publication should it be accepted. Please note that figures must have been originally created at 300dpi or higher. Do not manually increase the resolution of your files. For instructions on how to properly obtain high quality images, please review our Figure Guidelines, with examples at: http://journals.plos.org/plosntds/s/figures

While revising your submission, please upload your figure files to the Preflight Analysis and Conversion Engine (PACE) digital diagnostic tool, https://pacev2.apexcovantage.com/ PACE helps ensure that figures meet PLOS requirements. To use PACE, you must first register as a user. Then, login and navigate to the UPLOAD tab, where you will find detailed instructions on how to use the tool. If you encounter any issues or have any questions when using PACE, please email us at figures@plos.org.

We hope to receive your revised manuscript by Jan 13 2020 11:59PM. If you anticipate any delay in its return, we ask that you let us know the expected resubmission date by replying to this email.

To submit your revised files, please log in to https://www.editorialmanager.com/pntd/

Sincerely,

Andrew S. Azman

Deputy Editor

Reviewer's Responses to Questions

**Key Review Criteria Required for Acceptance?**

**Methods**

-Are the objectives of the study clearly articulated with a clear testable hypothesis stated?

-Is the study design appropriate to address the stated objectives?

-Is the population clearly described and appropriate for the hypothesis being tested?

-Is the sample size sufficient to ensure adequate power to address the hypothesis being tested?

-Were correct statistical analysis used to support conclusions?

-Are there concerns about ethical or regulatory requirements being met?

Reviewer #1: I think the revised manuscript is a lot easier to read, with more streamlined results and a clearer message. The new supporting Tables and Figures are helpful.

Reviewer #2: (No Response)

Reviewer #3: (No Response)

**Results**

-Does the analysis presented match the analysis plan?

-Are the results clearly and completely presented?

-Are the figures (Tables, Images) of sufficient quality for clarity?

Reviewer #1: (No Response)

Reviewer #2: (No Response)

Reviewer #3: (No Response)

**Conclusions**

-Are the conclusions supported by the data presented?

-Are the limitations of analysis clearly described?

-Do the authors discuss how these data can be helpful to advance our understanding of the topic under study?

-Is public health relevance addressed?

Reviewer #1: (No Response)

Reviewer #2: (No Response)

Reviewer #3: (No Response)

**Editorial and Data Presentation Modifications?**

Reviewer #1: In the last paragraph of the Introduction (lines 96-101), the mathematical modeling is mentioned but not the statistical analyses. It would be good to mention both approaches since they address different parts of your objectives.

I can't see a caption in the Table S4 doc file.

Caption for Table S7 refers to "each city".

For the caption of Figure S5, mention what the dashed black line is.

Reviewer #2: (No Response)

Reviewer #3: (No Response)

**Summary and General Comments**

Reviewer #1: (No Response)

Reviewer #2: I could not find any point-by-point response from authors to reviewers' comments. 

However, I am quite satisfied with the additional elements and justifications provided in the revised manuscript. 

To my opinion, the new version of the manuscript would benefit from a few additional minor modifications.

Additional details should be added in the Methods section to describe the hierarchical regression analyses. It seems that Authors used mixed models with fixed and random effect but do not clearly explain it.

The result section is quite dense and it could thus be easier to read after including subheadings. 

The new Results section includes a few sentences that may be displaced in the Discussion (ex. lines 289-290; lines 299-302).

Reviewer #3: (No Response)

PLOS authors have the option to publish the peer review history of their article (what does this mean?). If published, this will include your full peer review and any attached files.

Reviewer #1: No

Reviewer #2: Yes: Stanislas Rebaudet

Reviewer #3: Yes: Elizabeth Lee

---

## [Decision Letter · Decision Letter 2]

10 Jan 2020

Dear Mrs. Phillips,

We are pleased to inform you that your manuscript, "Changes in historical typhoid transmission across 16 U.S. cities, 1889-1931: Quantifying the impact of investments in water and sewer infrastructures", has been editorially accepted for publication at PLOS Neglected Tropical Diseases.

Before your manuscript can be formally accepted and sent to production you will need to complete our formatting changes, which you will receive in a follow up email. Please note: your manuscript will not be scheduled for publication until you have made the required changes.

IMPORTANT NOTES

* Copyediting and Author Proofs: To ensure prompt publication, your manuscript will NOT be subject to detailed copyediting and you will NOT receive a typeset proof for review. The corresponding author will have one final opportunity to correct any errors when sent the requests mentioned above. Please review this version of your manuscript for any errors.

* If you or your institution will be preparing press materials for this manuscript, please inform our press team in advance at plosntds@plos.org. If you need to know your paper's publication date for media purposes, you must coordinate with our press team, and your manuscript will remain under a strict press embargo until the publication date and time. PLOS NTDs may choose to issue a press release for your article. If there is anything that the journal should know, please get in touch.

*Now that your manuscript has been provisionally accepted, please log into EM and update your profile. Go to http://www.editorialmanager.com/pntd, log in, and click on the "Update My Information" link at the top of the page. Please update your user information to ensure an efficient production and billing process.

*Note to LaTeX users only - Our staff will ask you to upload a TEX file in addition to the PDF before the paper can be sent to typesetting, so please carefully review our Latex Guidelines [http://www.plosntds.org/static/latexGuidelines.action] in the meantime.

Best regards,

Andrew S. Azman

Deputy Editor

Reviewer's Responses to Questions

**Key Review Criteria Required for Acceptance?**

**Methods**

-Are the objectives of the study clearly articulated with a clear testable hypothesis stated?

-Is the study design appropriate to address the stated objectives?

-Is the population clearly described and appropriate for the hypothesis being tested?

-Is the sample size sufficient to ensure adequate power to address the hypothesis being tested?

-Were correct statistical analysis used to support conclusions?

-Are there concerns about ethical or regulatory requirements being met?

Reviewer #3: Yes

**Results**

-Does the analysis presented match the analysis plan?

-Are the results clearly and completely presented?

-Are the figures (Tables, Images) of sufficient quality for clarity?

Reviewer #3: Yes

**Conclusions**

-Are the conclusions supported by the data presented?

-Are the limitations of analysis clearly described?

-Do the authors discuss how these data can be helpful to advance our understanding of the topic under study?

-Is public health relevance addressed?

Reviewer #3: Yes

**Editorial and Data Presentation Modifications?**

Reviewer #3: (No Response)

**Summary and General Comments**

Reviewer #3: The authors have addressed my comments in a sufficient manner.

PLOS authors have the option to publish the peer review history of their article (what does this mean?). If published, this will include your full peer review and any attached files.

Reviewer #3: Yes: Elizabeth Lee

---

## [Editor Report · Acceptance letter]

2 Mar 2020

Dear Mrs. Phillips,

We are delighted to inform you that your manuscript, "Changes in historical typhoid transmission across 16 U.S. cities, 1889-1931: Quantifying the impact of investments in water and sewer infrastructures," has been formally accepted for publication in PLOS Neglected Tropical Diseases.

Best regards,

Serap Aksoy

Editor-in-Chief

Shaden Kamhawi

Editor-in-Chief
